# DISTRIBUTION-AWARE SYNERGISTIC EVOLUTION FOR FEW-SHOT DISCRIMINATION AND GENERATION

## ABSTRACT

Discrimination and generation are two distinct yet complementary paradigms in machine learning. In few-shot scenarios, prototype-based discriminative methods are better at estimating the *class center*, while generative models are better at modeling the *data variance*. To harness the strengths of both paradigms, we propose a synergistic evolution framework that allows discriminative and generative methodologies to cooperate in estimating feature distributions. For one thing, the discriminative model incorporates synthetic samples from the generative model to improve the estimation of feature covariance, especially when the available data is limited. For another, the generative model leverages calibrated class centers from the discriminative pathway as anchors to improve the semantic accuracy of the generated samples. In summary, our framework enables the discriminative model and the generative model to jointly develop and collaborate within a few-shot learning scenario, thereby enhancing both of their individual capabilities. Additionally, our design improves open-set learning by enhancing out-of-distribution detection through better covariance modeling in the discriminative space. Extensive experiments on the CUB200 and miniImageNet datasets demonstrate performance gains in few-shot class-incremental learning (FSCIL), few-shot image generation (FSIG), and open-set recognition (OSR) tasks.

## 1 INTRODUCTION

The divide between generative and discriminative paradigms has played a foundational role in shaping the evolution of computer vision methods. Discriminative models excel at learning decision boundaries through supervised training (Goodfellow et al., 2016), while generative models focus on capturing inner probabilistic distribution to synthesize diverse data samples (Oussidi & Elhassouny, 2018). Although these approaches pursue distinct objectives, their inherent complementarity has motivated persistent research into synergistic integration. A critical yet under-addressed challenge lies in establishing mutual reinforcement mechanisms that simultaneously enhance both generation and discrimination capabilities—particularly under few-shot learning scenarios where data scarcity amplifies the need for collaborative optimization.

Existing hybrid approaches predominantly employ adversarial frameworks (Goodfellow et al., 2020) or sequential pipelines (Antoniou et al., 2017), but often prioritize one paradigm over the other. For instance, GAN variants primarily optimize sample generation quality while treating discrimination as an auxiliary loss component (Goodfellow et al., 2020; Salimans et al., 2016). Conversely, data augmentation methods utilize generative models as mere suppliers of synthetic data, without benefiting from discriminative feedback (Antoniou et al., 2017).

To fill this research gap, we propose a **D**istribution-**A**ware **S**ynergistic **E**volution framework (**DASE**), that unifies generation and discrimination through joint optimization of class distributions. Our method establishes an evolvable feature space based on CLIP's representations (Radford et al., 2021): **In the first phase**, the generator and discriminator operate independently. The discriminator calibrates the mean representations of novel class distributions by referencing those of base classes and incorporating textual semantic information, while the generator additionally uses class distribution means as the condition of Stable Diffusion 1.5, to fine-tune the model with LoRA on few-shot samples to generate diverse images. **In the second phase**, there's a two-way knowledge transfer: the generated samples enrich covariance estimation for few-shot classes through controlled distribution expansion,

**enhancing the discriminator** by mitigating feature collapse in data-scarce regimes. Meanwhile, the calibrated distribution means obtained from the discriminative pathway act as semantic anchors, **guiding the generator** to ensure class fidelity in the generated outputs.

To summarize, our work introduces three key innovations:

- *Unified Framework.* We formalize a synergistic evolution mechanism in which discriminative calibration guides generative exploration, while the generated samples enhance the modeling of discriminative distributions. It supports the joint optimization of discriminative tasks, including Few-Shot Class-Incremental Learning and Open-Set Recognition, and the generative task Few-Shot Image Generation, within a unified framework.

- *Improvement for Discrimination.* We boost discrimination by using generated samples from the Generative Pathway to improve covariance estimation in few-shot classes, leading to more reliable few-shot class-incremental learning and improved out-of-distribution detection through better modeling of class distributions.

- *Improvement for Generation.* Leveraging the alignment between CLIP's visual and textual feature spaces, we implement a method to improve the class fidelity of generated samples using visual prototype features from the Discriminative pathway. The discriminator effectively guides the generator by appending the calibrated prototypes to the CLIP text encoder output in Stable Diffusion.

## 2 RELATED WORK

### 2.1 GENERATION AND DISCRIMINATION HYBRIDS

The interplay between generative and discriminative models has been extensively explored through two primary paradigms. One line of work centers on **adversarial learning**, particularly Generative Adversarial Networks (GANs) (Arjovsky et al., 2017; Goodfellow et al., 2020; Karras et al., 2020), which pit a generator against a discriminator in a min-max game to improve the quality of generated data. Variational Autoencoders (VAEs) (Kingma et al., 2013) and their derivatives also serve as foundational generative models that support downstream discrimination tasks by encoding structured latent representations. Another direction involves using generative models for **data augmentation**—generating synthetic samples or pseudo-labels to expand the training distribution and enhance the generalization of discriminative models (Antoniou et al., 2017).

Despite these efforts, most existing methods pursue only one aspect (generation or discrimination) and leave the other auxiliary, or lack true mutual reinforcement between the discriminative and generative pathways. In contrast, our work builds a distribution-aware collaboration mechanism that allows generated samples to refine covariance estimation in the discriminative space, while the calibrated means from the discriminative pathway guide the generator to maintain semantic fidelity.

### 2.2 FEW-SHOT CLASS INCREMENTAL LEARNING

Few-shot class-incremental learning (FSCIL) is a demanding continual learning scenario in which a model must incrementally learn new classes from very few examples, while not forgetting the previously learned classes. This setting faces two challenges: catastrophic forgetting of old classes(McCloskey & Cohen, 1989), and overfitting for novel classes(Wang et al., 2020).

With the emergence of powerful pre-trained visual backbones, prototype-based methods (Snell et al., 2017) have gained advantage and popularity, which effectively mitigate the forgetting issue by freezing the backbone and computing class-wise feature means as prototypes to preserve the base distribution. To address overfitting on novel classes, recent methods leverage semantic similarity between new and base classes to regularize prototype estimation (Akyürek et al., 2021; Cheraghian et al., 2021; Wang et al., 2023). Such semantic similarity is often derived from feature distances or external textual cues. For example, (Akyürek et al., 2021) constrains novel class weights within the base class subspace. (Cheraghian et al., 2021) introduces textual embeddings to construct semantically informed subspaces for prototype calibration. (Wang et al., 2023) exploits holistic similarity to align novel prototypes toward semantically close base classes.

More recently, vision-language pre-trained models have shown remarkable potential in FSCIL. Methods like LP-DiF (Huang et al., 2024) employ prompt tuning with distribution-based feature replay

to prevent catastrophic forgetting while adapting to new classes. PCL (Li et al., 2025) introduces prompt-based concept learning that generalizes conceptual knowledge from base to incremental sessions. Privilege (Park et al., 2024) leverages pre-trained vision and language transformers with specialized prompting functions and knowledge distillation to achieve superior few-shot incremental learning performance.

Inspired by these, we also use a frozen vision-language pre-trained model to provide feature prototypes and adopt text-guided fine-grained calibration(Zongyao et al., 2025) to better position novel class prototypes in the feature space, improving class separation and mitigating overfitting.

### 2.3 FEW-SHOT IMAGE GENERATION

Few-shot image generation (FSIG) aims to generate high-quality and diverse images using only a limited number of samples. Many existing works follow the domain adaptation paradigm proposed in (Wang et al., 2018), where a GAN pretrained on a large-scale source domain is transferred to a small target dataset. Our work follows another paradigm of Few-shot image generation, which involves training a model on a large number of images from seen categories and applying it to unseen categories with limited samples (Antoniou et al., 2017).

GAN-based approaches(Clouâtre & Demers, 2019; Hong et al., 2020; 2022) dominated early Few-shot image generation methods. ADAM(Zhao et al., 2022) is based on modulation, while preserving knowledge in the source generator that is critical for the adaptation task. RICK(Zhao et al., 2023) adopts a lightweight filter-pruning strategy, progressively removing the least important kernels.GenDA(Mondal et al., 2023) generates diverse target-domain samples by mapping target-domain samples to the latent space of a pre-trained source GAN using a latent-generation network during the inference stage.

With the advent of large-scale text-to-image diffusion models(Rombach et al., 2022), generative models have achieved significant improvements in image quality and have also driven the development of semantically guided approaches. Textual Inversion(Gal et al., 2022) introduces a pseudo-word embedding that is learned to represent a novel concept within the text-to-image generation framework. DreamBooth(Ruiz et al., 2023) enables personalized image generation by introducing a rare identifier as a category modifier and employing a class-specific prior preservation loss to maintain visual fidelity to the target class. While existing methods depend on introducing an identifier or a pseudo-word, our approach improves the quality of generated samples by optimizing the original text embedding inputs.

## 3 PRELIMINARIES

The modeling of class-conditional feature distributions faces challenges due to data scarcity and the high dimensionality of modern vision models. Under the standard FSCIL paradigm (Tao et al., 2020), a novel class $c$ emerges sequentially with only $K(K \leq 5)$ samples per category, introducing semantic bias for distribution mean and instability for covariance matrix estimation. Since the empirical covariance estimate $\Sigma \in \mathbb{R}^{d \times d}$ requires at least $d+1$ samples for full-rank estimation, and the CLIP's ViT-L backbone produces $d = 768$-dimensional features, 5-shot learning leaves the covariance matrix severely underdetermined (rank $\leq 5$ vs. required 768). This forces practical implementations to adopt diagonal approximations $\mathrm{diag}(\sigma_1^2, ..., \sigma_d^2)$, sacrificing cross-dimensional correlations but maintaining numerical stability.

Let $\mathcal{D}_b$ denote base classes with abundant samples and $\mathcal{D}_n$ represent few-shot novel classes. Given CLIP's demonstrated cross-modal alignment capabilities, we extract visual features $\mathbf{v}_i \in \mathbb{R}^d$ using its ViT-L backbone. For each class $c$, we model its feature distribution as $\mathcal{N}(\mu_c, \Sigma_c)$, where the class mean $\mu_c$ and diagonal covariance $\Sigma_c$ are computed as:

$$\mu_c = \frac{1}{|\mathcal{X}_c|} \sum_{\mathbf{x}_i \in \mathcal{X}_c} \mathbf{v}_i, \quad \Sigma_c = \mathrm{diag}\left(\frac{1}{|\mathcal{X}_c| - 1} \sum_{\mathbf{x}_i \in \mathcal{X}_c} (\mathbf{v}_i - \mu_c)^2\right) \tag{1}$$

where $\mathcal{X}_c$ denotes the sample set of class $c$. This probabilistic modeling supports both discriminative and generative learning scenarios, which share a common reliance on accurate distribution estimation.

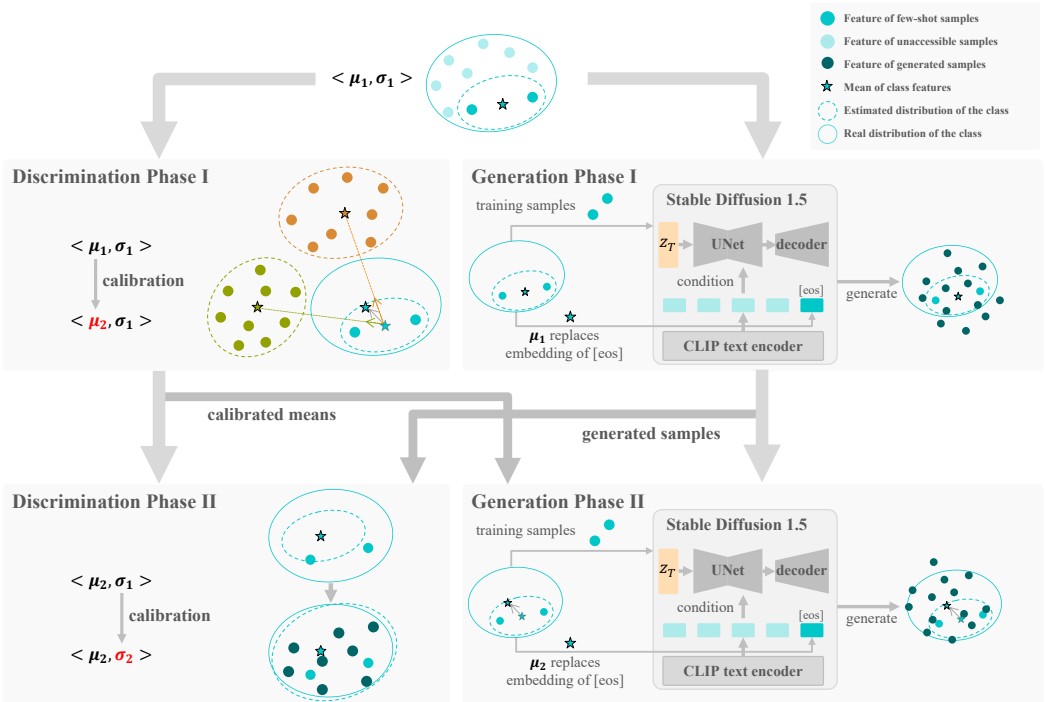

Figure 1: Overall architecture of our framework. Initially the scarce samples (only two) leads to biased mean estimation and unreliable covariance calculation. **(a) Discriminative pathway**: **In Phase I**, the distribution mean $\mu_1$ is improved by referencing base classes with abundant samples. $\mu_1$ is pulled closer to distribution mean of semantically similar base classes (orange and green). **In Phase II**, abundant generated samples from *Generative Pathway* are used for covariance estimation, enhancing the stability of covariance estimation. **(b) Generative pathway**: **In Phase I**, distribution mean $\mu_1$ (i.e. class prototype) is provided as a semantic anchor for the generative model. **In Phase II**, the further calibrated distribution mean $\mu_2$ from the *Discriminative Pathway* is used to guide the generator to better ensure class fidelity for synthetic samples.

For discriminative tasks, including **Few-Shot Class-Incremental Learning (FSCIL)** and **Open-Set Recognition (OSR)**, estimating the likelihood of a test sample under the modeled class distribution is essential. The probability density of a test feature vector $\mathbf{v}_*$ under class $c$ is given by:

$$p(\mathbf{v}_* \mid c) = \frac{1}{\sqrt{(2\pi)^d |\boldsymbol{\Sigma}_c|}} \exp\left(-\frac{1}{2}(\mathbf{v}_* - \boldsymbol{\mu}_c)^\top \boldsymbol{\Sigma}_c^{-1}(\mathbf{v}_* - \boldsymbol{\mu}_c)\right). \tag{2}$$

In FSCIL, a test sample is classified into the class with the highest probability density. In OSR, a sample is detected as out-of-distribution if its maximum density across all known classes falls below a predefined threshold.

For the generative task of **Few-Shot Image Generation (FSIG)**, the class mean $\boldsymbol{\mu}_c$ serves as a semantic anchor to condition the generation process. Incorporating $\boldsymbol{\mu}_c$ into the generative model—detailed in subsequent sections—enhances the semantic accuracy of the synthesized images.

## 4 FRAMEWORK

To effectively address the challenges of biased mean estimation and unreliable covariance calculation, we propose a **Distribution-Aware Synergistic Evolution (DASE)** framework that establishes a continuous feedback loop between discriminative and generative pathways. The core idea is to enable joint refinement of class distribution parameters by allowing each pathway to improve the other. Specifically, the discriminative model leverages generated samples to enhance covariance

estimation, while the generative model uses calibrated class prototypes from the discriminator to improve semantic fidelity.

The DASE framework operates on shared feature distributions derived from CLIP's visual encoder, ensuring alignment between visual and textual semantics. As illustrated in Figure 1, the framework consists of two parallel pathways—discriminative and generative—each undergoing a two-stage evolutionary process. Below, we detail the design and rationale of each component.

## 4.1 DISCRIMINATIVE PATHWAY

The discriminative pathway addresses the dual challenges of biased mean estimation and unreliable covariance calculation through a two-stage evolutionary process. It aims to accurately model class-conditional feature distributions under few-shot constraints. It evolves through two phases—calibration with semantic priors and refinement with generated samples—each designed to incrementally improve distribution estimates.

**Discrimination Phase I: Calibration with Semantic Priors**

In the initial calibration stage, we mitigate the bias in few-shot novel class means $\boldsymbol{\mu}_c$ by referring to the less biased base class means $\{\boldsymbol{\mu}_{b,b\in\mathcal{D}_b}\}$ with abundant samples. Specifically, wo adopt a calibration function (Zongyao et al., 2025)(denoted as $\mathcal{C}(\cdot)$), which utilizes textual semantic relations between novel and base classes for weighted calibration:

$$\boldsymbol{\mu}_c' = \mathcal{C}(\boldsymbol{\mu}_c, \mathcal{T}(c), \{\boldsymbol{\mu}_{b,b\in\mathcal{D}_b}\}), \tag{3}$$

where $\mathcal{T}$ denotes the CLIP's text encoder. This calibration reduces the prototype displacement caused by few-shot samples while preserving class distinctiveness. For covariance estimation, we introduce a regularized estimator that combines the empirical diagonal variance with base class statistics:

$$\boldsymbol{\Sigma}_c' = \underbrace{\lambda \cdot \boldsymbol{\Sigma}_c}_{\text{Few-shot estimate}} + \underbrace{(1-\lambda) \cdot \overline{\boldsymbol{\Sigma}}_b}_{\text{Base prior}} + \eta\mathbf{I}, \tag{4}$$

where $\overline{\boldsymbol{\Sigma}}_b$ is the average covariance of base classes, $\lambda$ controls base-novel knowledge transfer, and $\eta$ provides numerical stability.

**Discrimination Phase II: Calibration with Generated Samples**

In the refinement stage, synthetic samples from the generative pathway enable better covariance estimation. We first filter generated samples $\tilde{\mathcal{X}}_c$ using the discriminative model's confidence scores, retaining only samples with $p(y = c|\mathbf{x}) > \tau$. The diagonal covariance $\boldsymbol{\Sigma}_c$ then becomes:

$$\boldsymbol{\Sigma}_c' = \lambda \cdot \frac{w_r \sum_{\mathbf{x}_i \in \mathcal{X}_c}(\mathbf{v}_i - \boldsymbol{\mu}_c)^2 + w_g \sum_{\mathbf{x}_j \in \tilde{\mathcal{X}}_c}(\mathbf{v}_j - \boldsymbol{\mu}_c)^2}{w_r|\mathcal{X}_c| + w_g|\tilde{\mathcal{X}}_c|} + (1-\lambda) \cdot \overline{\boldsymbol{\Sigma}}_b + \eta\mathbf{I}, \tag{5}$$

where $w_r$ and $w_g$ balance contributions of real and generated samples.

## 4.2 GENERATIVE PATHWAY

The generative pathway aims to produce high-fidelity images for novel classes using only a few examples. It also operates in two phases: initial generation conditioned on visual prototypes, and refined generation using calibrated anchors from the discriminator.

**Generation Phase I: Text Embedding Enhanced by Visual Prototypes**

Text-to-image models like Stable Diffusion rely on textual prompts, which may not fully capture the visual characteristics of few-shot classes. By injecting visual prototype features into the text embedding, we align the generative process with actual visual semantics, improving class consistency.

In the Stable Diffusion pipeline, the input natural language prompt is first tokenized by the CLIP tokenizer and then encoded by the CLIP text encoder into a fixed-length sequence of embeddings:

$$tokens = [BOS,\ s_1,\ s_2,\ \ldots,\ s_m,\ EOS,\ PAD,\ \ldots], \tag{6}$$

$$T = \mathcal{T}(tokens) = [\boldsymbol{t_{BOS}}, \boldsymbol{t_1}, \boldsymbol{t_2}, \ldots, \boldsymbol{t_m}, \boldsymbol{t_{EOS}}, \boldsymbol{t_{PAD}}, \ldots], \qquad (7)$$

where EOS stands for *End Of Sentence*. Here, EOS produces an embedding $t_{EOS}$ that summarizes the global semantics of the entire prompt. All embeddings corresponding to EOS and PAD tokens are substituted with the category prototype $\boldsymbol{\mu_c}$, which is computed as the mean feature vector of all images in the category extracted by the CLIP image encoder:

$$T' = [\boldsymbol{t_{BOS}}, \boldsymbol{t_1}, \boldsymbol{t_2}, \ldots, \boldsymbol{t_m}, \boldsymbol{\mu_c}, \boldsymbol{\mu_c}, \ldots]. \qquad (8)$$

This substitution injects visual semantics directly into the generation process, leveraging CLIP's aligned cross-modal space to guide the diffusion model toward class-relevant outputs.

We fine-tune a pretrained text-to-image diffusion model using low-rank adaptation (LoRA). During fine-tuning, the original model parameters $\theta_0$ remain frozen, and only the LoRA parameters $\phi$ are updated. The denoising network with LoRA adaptation is denoted by $\epsilon_{\theta_0,\phi}$. The fine-tuning objective follows the standard noise-prediction loss used in diffusion models:

$$\mathcal{L} = \mathbb{E}_{(x,t),\,\epsilon,\,s} \left[ \left\| \epsilon - \epsilon_{\theta_0,\phi}(x_s, \, s, \, T') \right\|_2^2 \right]. \qquad (9)$$

Here, $x_s$ is the noisy sample constructed at timestep $s$, $\epsilon$ is Gaussian noise, and $T'$ denotes the text embedding produced by our custom text encoder. In this formulation, $\theta_0$ represents the frozen pretrained diffusion parameters, while $\phi$ denotes the trainable low-rank matrices introduced by LoRA.

**Generation Phase II: Calibrated anchors from discriminative pathway**

To further enhance generation fidelity, we refine the raw category prototype using fine-grained prototypes obtained from the discriminative pathway. The final embedding sequence becomes:

$$T' = [\boldsymbol{t_{BOS}}, \boldsymbol{t_1}, \boldsymbol{t_2}, \ldots, \boldsymbol{t_m}, \boldsymbol{\mu'_c}, \boldsymbol{\mu'_c}, \ldots], \qquad (10)$$

where $\boldsymbol{\mu'_c}$ denotes the calibrated category prototype.

By anchoring the diffusion process on this calibrated semantic prototype, the model is guided to produce higher-fidelity images that more accurately reflect the target class.

This framework effectively unifies discrimination and generation within a few-shot learning scenario, enabling mutual reinforcement and continuous improvement in both tasks.

## 5 EXPERIMENTS

This section first introduces the datasets, evaluation metrics, and implementation details used in our experiments on discriminative tasks, including FSCIL and OSR, and generative task FSIG. It then presents comparisons with existing and baseline methods, and concludes with ablation studies validating the effectiveness of our approach.

### 5.1 EXPERIMENT SETTINGS

**Datasets** We evaluate our method on two widely used benchmarks: CUB200 (Wah et al., 2011)and miniImageNet(Vinyals et al., 2016), following the FSCIL settings of previous works (Tao et al., 2020; Wang et al., 2023). CUB200 contains 200 bird species, which are split into 11 sessions. The base session consists of the first 100 classes as base classes, and the remaining 100 classes, used as novel classes, are split into 10 incremental sessions, each introducing 10 new classes with 5 images per class (10-way 5-shot). miniImageNet comprises 60,000 images selected from ImageNet (Russakovsky et al., 2015), containing 100 classes with 600 images per class. Each image is sized at $84 \times 84$ pixels. We use 60 classes for the base session and 40 for incremental sessions. The 40 novel classes are introduced over 8 sessions, each adding 5 new classes with 5 examples per class (5-way 5-shot).

**Evaluation metrics** We adopt multiple metrics to evaluate FSCIL, OSR, and FSIG performance. For FSCIL, session accuracy measures the model's performance on the seen classes up to this session on the test set. $A_{avg}$ computes the mean accuracy from the base session to the final session, reflecting the model's overall generalization across seen and unseen classes. $A_{last}$ denotes the accuracy achieved in the final session, reflecting the model's classification precision for all learned categories upon completing continual training. For OOD detection, the AUROC (Area Under the Receiver

Operating Characteristic curve) is utilized to measure the overall separability between ID and OOD distributions (higher values indicate better discrimination). The evaluation of OSR is conducted on each incremental session (except the final session). We treat the novel classes from the last session of the dataset as OOD categories and the seen classes of each session as ID categories, testing the model's open-set recognition capability. For generation tasks, we report the Fréchet Inception Distance (FID)(Heusel et al., 2017), which evaluates the quality and diversity of generated images by comparing their feature distribution to real images. A lower FID score indicates more realistic and diverse generations.

**Implementation Details** Our framework is implemented in PyTorch with CLIP-ViT/L-16 as the backbone. Both the discriminative pathway and generative pathway utilize the identical pre-trained CLIP model, which remains frozen throughout the entire FSCIL and FSIG processes. For the discriminative pathway, 768-dimensional visual and textual features are extracted from CLIP's aligned embedding space, and the distribution of each class is modeled using Gaussian distributions. For the generative pathway, LoRA training is performed with a rank of 128 over 400 steps, an initial learning rate of $1 \times 10^{-4}$, and a cosine learning rate decay schedule. During image generation, the diffusion model combined with LoRA employs 50 sampling steps and a classifier-free guidance scale of 6.5. To improve generalization to novel classes and prevent overfitting, we adopt class-specific prior preservation loss during LoRA training on the CUB-200 dataset. This loss jointly optimizes the model on both the novel-class images and a regularization dataset, following the DreamBooth paradigm. Specifically, the textual prompts for dataset images are formatted as 'a photo of a [class name] bird, [description]', whereas the regularization images are generated from the base Stable Diffusion v1.5 model using the prompt 'a photo of a bird'. The regularization set is used in a 1:1 ratio relative to the training data.

## 5.2 COMPARISON WITH EXISTING METHODS

Table 1: Overall result of comparative experiments on Few-shot class-incremental learning. **Bold** indicates best performance.

| Method | Backbone | miniImageNet | | CUB200 | |
|---|---|---|---|---|---|
| | | Alast(%) | Aavg(%) | Alast(%) | Aavg(%) |
| LP-DiF (Huang et al., 2024) | CLIP-B | 91.68 | 93.76 | 68.53 | 74.03 |
| PCL (Li et al., 2025) | CLIP-B | 92.28 | 94.02 | 68.31 | 75.38 |
| PriVilege (Park et al., 2024) | CLIP-B | 91.84 | 93.89 | 68.42 | 75.12 |
| LP-DiF (Huang et al., 2024) | CLIP-L | 93.46 | 94.67 | 72.57 | 77.57 |
| PCL (Li et al., 2025) | CLIP-L | 93.26 | 94.53 | 72.48 | 77.91 |
| PriVilege (Park et al., 2024) | CLIP-L | 92.78 | 94.45 | 76.36 | 79.13 |
| Baseline (Snell et al., 2017) | CLIP-L | 91.02 | 92.34 | 77.24 | 79.58 |
| DASE(Phase I) | CLIP-L | 92.60 | 94.33 | 78.51 | 80.49 |
| DASE(Phase II) | CLIP-L | **93.52** | **94.71** | **80.45** | **82.36** |

Table 2: Overall result of comparative experiments on Open-Set Recognition (AUROC %). **Bold** indicates best performance.

| Method | Backbone | miniImageNet | CUB200 |
|---|---|---|---|
| Baseline | CLIP-L | 50.19 | 42.77 |
| DASE(Phase I) | CLIP-L | 54.23 | 45.78 |
| DASE(Phase II) | CLIP-L | **55.34** | **47.08** |

**Few-shot class-incremental learning** Table 1 presents the results of our method compared with recent works on FSCIL. To align with the CLIP-L used in table Diffusion v1.5's text encoder, our image encoder also employs CLIP-L. To ensure fair comparison, all compared methods were tested with their backbone networks replaced by CLIP-L. The baseline method extends the prototype network by incorporating modeling of class distributions. Despite its simplicity, this baseline proves highly effective—particularly on CUB200, where it surpasses several recent works.

Our approach further elevates performance, achieving $A_{avg}$ improvements of **2.37%** on miniImageNet and **2.78%** on CUB200 over the baseline. These gains validate the efficacy of the proposed DASE framework. In Phase I, we calibrate the mean of class distributions. The resulting improvement over the baseline underscores the importance of distribution-aware refinement. In Phase II, synthetic samples from the generative pathway further enhance covariance estimation. The additional gain in this stage confirms the contribution of generative-discriminative synergy. With the aid of generated samples, our method outperforms existing alternatives.

**Open-Set Recognition** The open-set recognition results are summarized in Table 2. The evaluation derives from FSCIL setting (Tao et al., 2020) under the assumption of disjoint class sets across sessions. Specifically, at each incremental session—after the model has been updated with the current session's classes—the test samples from all classes seen up to that session are regarded as in-distribution (ID) data, while samples from the last session are treated as out-of-distribution (OOD). The task is to determine whether a test sample is ID or OOD based on the confidence scores derived from the estimated class distributions at each session except the last.

Using a few-shot estimated class distribution without calibration as the baseline, our method consistently outperforms it, improving AUROC by **5.15%** on miniImageNet and **4.31%** on CUB200. The Phase I improvement stems from calibrated class distribution means, and the further performance gain in Phase II highlights the critical role of generated samples: by enriching covariance estimation, they yield more robust and complete class-conditional distributions. The generated samples help explore the real class boundaries and strengthen the model's ability to distinguish known from unknown categories. The overall superior performance confirms that our DASE framework enhances open-set recognition as a direct outcome of improved distribution modeling.

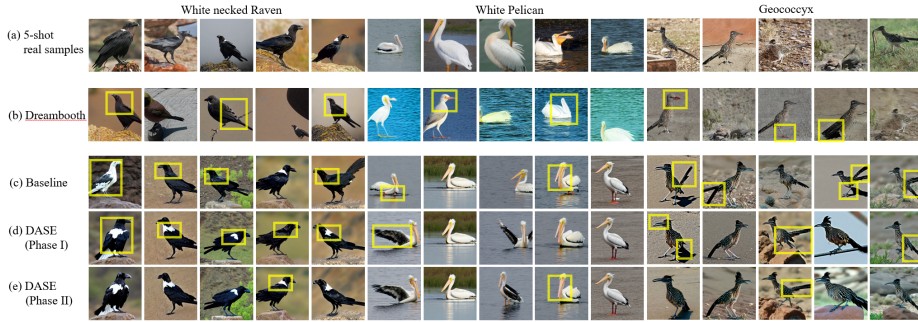

Figure 2: Qualitative comparison of image generation under different conditions. (a) 5-shot real training samples from CUB200. (b) Generation by the Dreambooth method (c) Baseline generation without class guidance. (d) Generation with mean feature prototypes. (e) Generation with calibrated feature prototypes. Specifically, DASE (Phase I) rectifies the feather color inaccuracies present in ravens in the baseline, and DASE (Phase 2) further recovers the missing details in the white-necked regions. Meanwhile, our method gradually captures the distinctive beak morphology of the White Pelican. Semantic alignment and visual quality improve progressively from (c) to (e).

Table 3: Overall result of comparative experiments on Few-Shot Image Generation. **Bold** indicates best performance.

| Method | Backbone | miniImageNet | | CUB200 | |
|---|---|---|---|---|---|
| | | ACC(%) | FID | ACC(%) | FID |
| Baseline | CLIP-L | 82.96 | 60.42 | 77.82 | 20.66 |
| DASE(Phase I) | CLIP-L | 83.49 | 51.70 | 77.98 | 20.44 |
| DASE(Phase II) | CLIP-L | **84.63** | **50.41** | **78.12** | **19.44** |

**Few-shot Image Generation** Table 3 demonstrates the improvements of our method on the FSIG task when using visual prototypes from the *Discriminative Pathway* as semantic anchors. Figure 2 provides a qualitative comparison of the contrasted methods on the CUB200 dataset. The baseline method fine-tunes the generator using only the training samples from the dataset. Our Phase I method

employs the class visual prototype as the semantic anchor for the generator, while our Phase II method utilizes the further calibrated and semantically more accurate class prototype as the semantic anchor.

The generated images of Phase I outperform those of the baseline in both class accuracy (ACC) and generation quality (FID), demonstrating the contribution of visual prototypes to sample generation. Specifically, ACC is computed by training a ground-truth classifier with all images of the corresponding class and then evaluating the class correctness of generated images. FID is calculated based on the distributional difference between generated samples and real samples from the corresponding dataset, serving as a measure of generation quality and fidelity. In Phase II, further improvements are observed, where the FID decreases by **10.01**(a 16.6% reduction) on miniImageNet and **1.22**(a 5.9% reduction) on CUB200 compared with the baseline, while the ACC also exhibits consistent gains. The additional performance gains of Phase II validate how prototype calibration in the Discriminative Pathway benefits the generative model, thereby further confirming that our framework achieves synergistic evolution between generation and discrimination.

### 5.3 ABLATION STUDY

Table 4: Ablation study of discriminative tasks (FSCIL and OSR) on the CUB200 dataset with different discriminative and generative components of our framework.

| Discriminative Prototypes | Generative samples | $A_{last}(\%)$ | $A_{avg}(\%)$ | AUROC(%) |
|---|---|---|---|---|
| Class Mean | none | 77.03 | 79.56 | – |
| Class Distribution | none | 77.24 | 79.58 | 34.57 |
| Calibrated Class Distribution | none | 78.51 | 80.49 | 42.77 |
| Calibrated Class Distribution | Simple Data Augmentation | 78.23 | 80.54 | 42.68 |
| Calibrated Class Distribution | Generative Model(Baseline) | 79.51 | 81.58 | 45.78 |
| Calibrated Class Distribution | Generative Model(Phase I) | 80.45 | 82.36 | 47.08 |

To evaluate the contribution of each component in our discriminative and generative framework, we conduct a series of ablations by progressively incorporating different strategies for modeling discriminative prototypes and leveraging generated samples to refine covariance estimation. Starting from the baseline method (Snell et al., 2017), which uses the class mean as the prototype, we incrementally introduce the following elements: class distribution estimation, class distribution calibration, covariance refinement using samples from simple data augmentation (random flipping and cropping), covariance refinement using generated samples from the baseline generative model (as in Table 3), and finally, our full method where the generative model employs calibrated prototypes from the discriminative pathway to produce augmented samples.

The results in Table 4 show that using simple data augmentation for covariance calibration brings negligible gains in discrimination and even degrades OSR performance. This suggests that low-level transformations like flipping and cropping fail to enrich scene diversity or semantic content under limited data, thus offering little support for exploring class distribution boundaries. In contrast, using samples from the generative model significantly improves discriminative performance. Moreover, generated samples guided by visual prototypes as semantic anchors lead to greater improvement than those from the baseline generator, as they exhibit higher semantic alignment and thus better calibrate the class distribution. Overall, the progressive inclusion of each component (except Simple Data Augmentation) consistently enhances both discriminative accuracy on FSCIL and OSR, validating the effectiveness of our design choices.

## 6 CONCLUSION

This paper proposes a distribution-aware synergistic evolution framework that unifies discriminative and generative paradigms for few-shot tasks. By establishing a bidirectional feedback loop, our method enables discriminative models to leverage synthetic samples for robust covariance estimation, while generative models utilize calibrated class prototypes from the discriminative pathway to enhance semantic fidelity in image generation. Our work bridges the gap between discrimination and generation under few-shot constraints, offering a scalable solution for class-incremental learning, open-set recognition, and few-shot image generation.

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
