# OpenReview forum: "Distribution-Aware Synergistic Evolution for Few-shot Discrimination and Generation"
_ICLR.cc/2026/Conference — Submitted to ICLR 2026_

### Official Review · Reviewer_7Td4 · 2025-10-26

**Soundness:** 3
**Presentation:** 2
**Contribution:** 3
**Rating:** 6
**Confidence:** 4

**Summary:**

1.  Uses generative samples to refine covariance estimation, improving few-shot class-incremental learning (FSCIL) and open-set recognition (OSR).

2. Leverages calibrated class prototypes from the discriminator to enhance semantic information in few-shot image generation

3. Enables mutual reinforcement between discrimination and generation, outperforming baselines on CUB200 and miniImageNet

**Strengths:**

1. Novel bidirectional interaction between discriminative and generative models

2. Introduces calibrated prototypes for generation and covariance refinement via synthetic samples

3. Rigorous experiments on FSCIL, OSR, and FSIG, with ablation studies validating each component

4. Unifies traditionally disjoint paradigms, enabling applications like continual learning with generative feedback

**Weaknesses:**

1. Experiments limited to Stable Diffusion 1.5 + LoRA. Larger diffusion models or non-CLIP backbones are untested.

2. Diagonal covariance ignores cross-feature correlations. A low-rank or sparse approximation could better capture structure.

3. No analysis of adversarial robustness such as distribution shifts.

**Questions:**

1. How does DASE perform with larger diffusion models or non-CLIP backbones?

2. Could structured (e.g., block-diagonal) covariance improve discrimination without overfitting?

3. How does OSR performance degrade under domain shifts (e.g., CUB200 → iNaturalist)?

4. Is there any theoretical support for the arguments presented in the abstract? "Generally, discriminative models are better at estimating the class center, while generative models are better at modeling the data variance."

---

> ### Author Response · Authors · 2025-11-28
>
> We appreciate the reviewer's constructive comments and have provided a point-by-point response to each, as detailed below.
>
> > 1. Diagonal covariance ignores cross-feature correlations. A low-rank or sparse approximation could better capture structure.
> > 2. Could structured (e.g., block-diagonal) covariance improve discrimination without overfitting?
>
> We thank the reviewers for these insightful suggestions. To thoroughly investigate this aspect, we conducted extensive ablation studies comparing full covariance, low-rank approximation (Probabilistic PCA retaining 3 principal components and block-diagonal covariance using 2×2 blocks), and diagonal covariance approach. The results are summarized below:
>
>
> | Method         | Backbone | Full Covariance |           |      | Probabilistic PCA |           |      | Block-Diagonal  Covariance |           |      | Diagonal Covariance |           |
> | -------------- | -------- | --------------- | --------- | ---- | ----------------- | --------- | ---- | -------------------------- | --------- | ---- | ------------------- | --------- |
> |                |          | $A_{last}$      | $A_{avg}$ |      | $A_{last}$        | $A_{avg}$ |      | $A_{last}$                 | $A_{avg}$ |      | $A_{last}$          | $A_{avg}$ |
> | Baseline       | CLIP-L   | 37.138          | 51.66     |      | 77.12             | 79.47     |      | 76.96                      | 79.29     |      | 77.24               | 79.58     |
> | Ours(Phase I)  | CLIP-L   | 37.138          | 51.66     |      | 78.54             | 80.52     |      | 78.35                      | 80.36     |      | 78.51               | 80.49     |
> | Ours(Phase II) | CLIP-L   | 37.138          | 51.66     |      | 80.38             | 82.25     |      | 80.21                      | 82.04     |      | 80.45               | 82.36     |
>
>
> The results demonstrate that under the extremely low-data regime (5 shots per class), both diagonal covariance and low-rank approximation (PPCA and block-diagonal) maintain comparable classification performance, while the full covariance approach fails completely. This can be attributed to several factors: both diagonal and low-rank approximations provide relatively stable distribution modeling in data-scarce conditions. However, even the low-rank approximations struggles to effectively capture meaningful cross-feature correlations with such limited samples, resulting in minimal performance gains over the simpler diagonal approach. Conversely, the full covariance matrix becomes severely underdetermined and numerically unstable in this setting, leading to near-zero classification accuracy for novel few-shot classes.
>
> These findings validate our practical choice of diagonal covariance as a compromise that balances model expressiveness with estimation stability given the severe data constraints.
>
>
>
> > 3. No analysis of adversarial robustness such as distribution shifts. How does OSR performance degrade under domain shifts (e.g., CUB200 → iNaturalist)?
>
> We appreciate this suggestion regarding domain shift robustness. We have evaluated the OSR performance under cross-domain settings where the model is continually learned on CUB200 (5-shot samples), and tested using different OOD datasets (iNaturalist, CIFAR-100). The results are as follows (AUROC%):
>
> | Method         | Backbone | CUB200→ CUB200 | CUB200→ iNaturalist | CUB200→ cifar100 |
> | -------------- | -------- | -------------- | ------------------- | ---------------- |
> | Baseline       | CLIP-L   | 42.77          | 67.54               | 58.96            |
> | Ours(Phase I)  | CLIP-L   | 45.78          | 70.93               | 61.24            |
> | Ours(Phase II) | CLIP-L   | 47.08          | 72.56               | 64.31            |
>
> Our method demonstrates consistent improvements in OSR performance under domain shift scenarios. The performance gain is attributed to our distribution-aware framework, where the generated samples help explore a more robust and generalized feature boundary. This enriched covariance estimation, guided by calibrated prototypes, makes the model less reliant on domain-specific features and more sensitive to semantic discrepancies, thereby enhancing its ability to detect OOD samples even when the domain shifts.

---

> > ### Author Response · Authors · 2025-11-28
> >
> > > 4. Experiments limited to Stable Diffusion 1.5 + LoRA. Larger diffusion models or non-CLIP backbones are untested.
> > >
> > > 5. How does DASE perform with larger diffusion models or non-CLIP backbones?
> >
> > We acknowledge this limitation in our current experimental scope. Our framework's generative-to-discriminative feedback loop is model-agnostic, but the specific bridge from discrimination to generation (via prototype injection) currently relies on the aligned vision-language feature space of CLIP. This design choice is pivotal for enabling the calibrated visual prototypes to effectively guide the text-conditioned generation in Stable Diffusion.
> >
> > We view it as an exciting avenue for future work to exploring adaptation to larger diffusion models and non-CLIP backbones, and believe the core synergistic concept of DASE can be extended beyond the current specific instantiation.
> >
> >
> >
> > > 6. Is there any theoretical support for the arguments presented in the abstract? "Generally, discriminative models are better at estimating the class center, while generative models are better at modeling the data variance."
> >
> > We thank the reviewer for this insightful comment and find the assertion in the abstract lacked sufficient contextualization. To clarify, our claim is specifically intended within the context of **few-shot learning scenarios**. Therefore, we have rephrased the statement in the manuscript to: "In few-shot scenarios, prototype-based discriminative methods are better at estimating the class center, while generative models are better at modeling the data variance."
> >
> > This revised formulation more accurately reflects the specific strengths of each model type, as evidenced by prior literature and our methodological design:
> >
> > **For Discriminative Models:**
> >
> > Discriminative models excel at learning decision boundaries by focusing on the conditional probability P(Y|X) and enhance **inter-class separability**.[1] In the context of prototype-based methods, this translates to a more accurate estimation of class centroids[2].  This principle is foundational to many few-shot learning algorithms, most notably **Prototypical Networks**[2], which explicitly leverage class centers for classification . The discriminative pathway in our framework is built directly upon this established principle.
> >
> > **For Generative Models:**
> >
> > Generative models aim to capture the underlying data distribution P(X) or P(X|Y). To generate realistic and diverse samples, they must capture the **intrinsic similarities among samples from the same class and model the complete data variance** [3]. This capability is empirically demonstrated by the success of various deep generative models, such as GANs and Diffusion Models, which are designed to learn complex, high-dimensional data distributions and have been shown to effectively **capture the feature distribution** of classes, even from limited data . Our use of a generative component to "expand" the estimated distribution for few-shot classes is designed to leverage this inherent capability of modeling detailed data variance.
> >
> > ---
> >
> > **References**
> >
> > 1. Hastie, Trevor. "The elements of statistical learning: data mining, inference, and prediction." (2009).
> > 2. Snell, Jake, Kevin Swersky, and Richard Zemel. "Prototypical networks for few-shot learning." *Advances in neural information processing systems* 30 (2017).
> > 3. Goodfellow, Ian J., et al. "Generative adversarial nets." *Advances in neural information processing systems* 27 (2014).

---

### Official Review · Reviewer_nbJQ · 2025-10-27

**Soundness:** 2
**Presentation:** 2
**Contribution:** 2
**Rating:** 4
**Confidence:** 4

**Summary:**

The paper proposes a unified framework called Distribution-Aware Synergistic Evolution (DASE) to integrate discriminative and generative learning for few-shot class-incremental learning (FSCIL), open-set recognition (OSR), and few-shot image generation (FSIG). It models each class as a Gaussian distribution in CLIP feature space and uses calibrated means and variances for both classification and generation. The approach consists of two phases: an initialization phase where visual prototypes are calibrated using base class statistics and text guidance, and a synergy phase where classifier and generator iteratively refine each other. Experiments on CUB-200 and miniImageNet demonstrate improvements across classification, out-of-distribution detection, and generative quality.

**Strengths:**

Presents a unified framework that leverages distribution-aware Gaussian modeling in CLIP feature space to support few-shot classification, open-set recognition, and image generation within a shared probabilistic structure.

**Weaknesses:**

1. The reliance on diagonal Gaussian assumptions, while practical for few-shot settings, may oversimplify class feature distributions.
2. The method depends heavily on heuristic filtering of synthetic data and text-guided prototype calibration, yet lacks a detailed ablation or sensitivity analysis to understand the robustness of these design choices.

**Questions:**

1. How sensitive is the overall performance to the quality of the synthetic samples selected during Phase II? Would the model degrade significantly if lower-quality generations were mistakenly included?
2. Have the authors considered or tested more expressive distribution models beyond diagonal Gaussians, such as full covariances, low-rank approximations, or mixture models? If so, what were the tradeoffs?
3. Could the text-guided calibration process introduce bias if semantic similarity does not align with visual similarity? How robust is the calibration step when text embeddings are noisy or ambiguous?
4. In the synergy phase, how often is feedback exchanged between the generator and classifier, and how does this frequency affect convergence and performance?
5. Does the method generalize well to non-fine-grained datasets or other domains (e.g., medical, synthetic imagery), or is it limited by assumptions baked into CLIP and Stable Diffusion’s pretraining?
6. Could the authors provide a more comprehensive ablation study that isolates the contributions of the synergy phase—specifically the generator-to-discriminator feedback via synthetic data and the discriminator-to-generator guidance via calibrated prototypes—across all three tasks (FSCIL, OSR, FSIG), as these components appear central to the claimed performance gains?

---

> ### Author Response · Authors · 2025-11-28
>
> We appreciate the reviewer's constructive comments and have provided a point-by-point response to each, as detailed below.
>
> > 1. The reliance on diagonal Gaussian assumptions, while practical for few-shot settings, may oversimplify class feature distributions.
> >
> > 2. Have the authors considered or tested more expressive distribution models beyond diagonal Gaussians, such as full covariances, low-rank approximations, or mixture models? If so, what were the tradeoffs?
>
> We thank the reviewers for these insightful suggestions. To thoroughly investigate this aspect, we conducted extensive ablation studies comparing full covariance, low-rank approximation (Probabilistic PCA retaining 3 principal components and block-diagonal covariance using 2×2 blocks), and diagonal covariance approach. The results are summarized below:
>
>
> | Method         | Backbone | Full Covariance |           |      | Probabilistic PCA |           |      | Block-Diagonal  Covariance |           |      | Diagonal Covariance |           |
> | -------------- | -------- | --------------- | --------- | ---- | ----------------- | --------- | ---- | -------------------------- | --------- | ---- | ------------------- | --------- |
> |                |          | $A_{last}$      | $A_{avg}$ |      | $A_{last}$        | $A_{avg}$ |      | $A_{last}$                 | $A_{avg}$ |      | $A_{last}$          | $A_{avg}$ |
> | Baseline       | CLIP-L   | 37.138          | 51.66     |      | 77.12             | 79.47     |      | 76.96                      | 79.29     |      | 77.24               | 79.58     |
> | Ours(Phase I)  | CLIP-L   | 37.138          | 51.66     |      | 78.54             | 80.52     |      | 78.35                      | 80.36     |      | 78.51               | 80.49     |
> | Ours(Phase II) | CLIP-L   | 37.138          | 51.66     |      | 80.38             | 82.25     |      | 80.21                      | 82.04     |      | 80.45               | 82.36     |
>
> The results demonstrate that under the extremely low-data regime (5 shots per class), both diagonal covariance and low-rank approximation (PPCA and block-diagonal) maintain comparable classification performance, while the full covariance approach fails completely. This can be attributed to several factors: both diagonal and low-rank approximations provide relatively stable distribution modeling in data-scarce conditions. However, even the low-rank approximations struggles to effectively capture meaningful cross-feature correlations with such limited samples, resulting in minimal performance gains over the simpler diagonal approach. Conversely, the full covariance matrix becomes severely underdetermined and numerically unstable in this setting, leading to near-zero classification accuracy for novel few-shot classes.
>
> These findings validate our practical choice of diagonal covariance as a compromise that balances model expressiveness with estimation stability given the severe data constraints.

---

> > ### Author Response · Authors · 2025-11-28
> >
> > > 3. The method depends heavily on heuristic filtering of synthetic data and text-guided prototype calibration, yet lacks a detailed ablation or sensitivity analysis to understand the robustness of these design choices.
> > >
> > > 4. How sensitive is the overall performance to the quality of the synthetic samples selected during Phase II? Would the model degrade significantly if lower-quality generations were mistakenly included?
> >
> > To address the sensitivity of synthetic sample quality, we conducted an analysis of the confidence threshold $\tau$ used for filtering generated samples in Discrimination Phase II:
> >
> > | $\tau$         | 0     | 0.25  | 0.5   | 0.75  | 1     |
> > | -------------- | ----- | ----- | ----- | ----- | ----- |
> > | Baseline       | 42.62 | 42.69 | 42.77 | 42.72 | 42.64 |
> > | Ours(Phase I)  | 45.66 | 45.72 | 45.78 | 45.74 | 45.68 |
> > | Ours(Phase II) | 46.97 | 47.04 | 47.08 | 47.05 | 47.01 |
> >
> > The results demonstrate that the optimal threshold is $\tau$=0.5, which effectively filters out low-confidence generated samples while retaining sufficient data for covariance refinement. Performance remains relatively stable around this optimum, showing our method's robustness to minor variations in sample selection. Moreover, $\tau$=0 causes performance degradation, confirming the importance of selective sample utilization.
> >
> > Regarding text-guided prototype calibration, our ablation study (Table 4) shows its consistent benefit:
> >
> > | Discriminative Prototypes     | Generative samples | $A_{last}$ | $A_{avg}$ |
> > | ----------------------------- | ------------------ | ---------- | --------- |
> > | Class Distribution            | none               | 77.24      | 79.58     |
> > | Calibrated Class Distribution | none               | 78.51      | 80.49     |
> >
> > As the calibration method itself is not our core contribution, we refer readers to [1] for detailed analysis of its mechanism and robustness.
> >
> > ---
> >
> > **References**
> >
> > [1] Xue Z, Kan M, Shan S, et al. Feature Decomposition-Recomposition in Large Vision-Language Model for Few-Shot Class-Incremental Learning. In: Proceedings of the IEEE/CVF International Conference on Computer Vision. 2025: 3153-3162.
> >
> >
> >
> > > 5. Could the text-guided calibration process introduce bias if semantic similarity does not align with visual similarity? How robust is the calibration step when text embeddings are noisy or ambiguous?
> >
> > **Regarding potential bias introduced by the text-guided calibration process:**
> >
> > The reviewer raises a valid concern. The calibration process could potentially introduce bias in specific domains where semantic similarity does not align well with visual similarity. However, it is important to emphasize that the text-guided calibration mechanism itself is also built upon CLIP's aligned vision-language feature space [1], where semantic similarity is designed to align with visual similarity across most common domains. This foundational alignment helps mitigate significant discrepancies in most practical scenarios.
> >
> > **Regarding robustness to noisy or ambiguous text embeddings:**
> >
> > The calibration step incorporates several design elements that ensure reasonable robustness. The weighting algorithm, which operates based on semantic similarity, includes mechanisms that prevent over-reliance on any single semantic source. Additionally, the calibration preserves a portion of the original prototype's weighting, providing a stabilizing effect against noisy or ambiguous embeddings. Furthermore, the text descriptions utilized in the calibration are generated by large language models based on class names, which produce semantically coherent and stable representations.
> >
> > ---
> >
> > **References**
> >
> > [1] Xue Z, Kan M, Shan S, et al. Feature Decomposition-Recomposition in Large Vision-Language Model for Few-Shot Class-Incremental Learning. In: Proceedings of the IEEE/CVF International Conference on Computer Vision. 2025: 3153-3162.

---

> > > ### Author Response · Authors · 2025-11-28
> > >
> > > > 6. In the synergy phase, how often is feedback exchanged between the generator and classifier, and how does this frequency affect convergence and performance?
> > >
> > > In our current framework, the feedback exchange between generator and classifier occurs only once in Phase II. We empirically found that additional iterations do not yield significant further improvements because: (1) the calibrated prototypes from the discriminator remain stable after the initial calibration, and (2) while higher-quality generated samples could theoretically provide marginal gains, the confidence-based filtering in Phase II already selects the most beneficial samples for covariance estimation. Given the computational cost of additional iterations and the diminishing returns observed, we adopted this single-exchange design as an effective balance between performance and efficiency.
> > >
> > >
> > >
> > > > 7. Does the method generalize well to non-fine-grained datasets or other domains (e.g., medical, synthetic imagery), or is it limited by assumptions baked into CLIP and Stable Diffusion’s pretraining?
> > >
> > > Our experiments on miniImageNet demonstrate the method's generalization capability beyond fine-grained domains (see Table 1). Additionally, we evaluated cross-domain OSR performance:
> > >
> > > | Method         | Backbone | CUB200→ CUB200 | CUB200→ iNaturalist | CUB200→ cifar100 |
> > > | -------------- | -------- | -------------- | ------------------- | ---------------- |
> > > | Baseline       | CLIP-L   | 42.77          | 67.54               | 58.96            |
> > > | Ours(Phase I)  | CLIP-L   | 45.78          | 70.93               | 61.24            |
> > > | Ours(Phase II) | CLIP-L   | 47.08          | 72.56               | 64.31            |
> > >
> > > Our method demonstrates consistent improvements in OSR performance under domain shift scenarios. The performance gain is attributed to our distribution-aware framework, where the generated samples help explore a more robust and generalized feature boundary. This enriched covariance estimation, guided by calibrated prototypes, makes the model less reliant on domain-specific features and more sensitive to semantic discrepancies, thereby enhancing its ability to detect OOD samples even when the domain shifts.
> > >
> > >
> > >
> > >
> > >
> > > > 8. Could the authors provide a more comprehensive ablation study that isolates the contributions of the synergy phase—specifically the generator-to-discriminator feedback via synthetic data and the discriminator-to-generator guidance via calibrated prototypes—across all three tasks (FSCIL, OSR, FSIG), as these components appear central to the claimed performance gains?
> > >
> > > We thank the reviewer for this suggestion. The generator-to-discriminator feedback corresponds to the transition from Discrimination Phase I to Phase II, while the discriminator-to-generator guidance corresponds to Generation Phase I to Phase II. Below we reorganize our results to explicitly isolate these contributions:
> > >
> > > **Generator-to-Discriminator Feedback (Synthetic Data Impact):**
> > >
> > > | Method                   | Backbone | miniImageNet |            |         | CUB200      |            |         |
> > > | ------------------------ | -------- | ------------ | ---------- | ------- | ----------- | ---------- | ------- |
> > > |                          |          | $A_{last}$↑  | $A_{avg}$↑ | AUROC ↑ | $A_{last}$↑ | $A_{avg}$↑ | AUROC ↑ |
> > > | Discrimination  Phase I  | CLIP-L   | 92.60        | 94.33      | 54.23   | 78.51       | 80.49      | 45.78   |
> > > | Discrimination  Phase II | CLIP-L   | 93.52        | 94.71      | 55.34   | 80.45       | 82.36      | 47.08   |
> > >
> > > **Discriminator-to-Generator Feedback (Calibrated Prototypes Impact):**
> > >
> > > | Method              | Backbone | miniImageNet |       | CUB200 |       |
> > > | ------------------- | -------- | ------------ | ----- | ------ | ----- |
> > > |                     |          | ACC ↑        | FID ↓ | ACC ↑  | FID ↓ |
> > > | Generation Phase I  | CLIP-L   | 83.49        | 51.70 | 77.98  | 20.44 |
> > > | Generation Phase II | CLIP-L   | 84.63        | 50.41 | 78.12  | 19.44 |
> > >
> > > These results clearly demonstrate that both feedback directions contribute significantly to performance gains across all three tasks: FSCIL ($A_{last}$, $A_{avg}$), OSR (AUROC), and FSIG (ACC, FID). The synergistic exchange enables continuous refinement, where each component enhances the other in a mutually beneficial cycle.

---

### Official Review · Reviewer_iTVM · 2025-10-31

**Soundness:** 2
**Presentation:** 2
**Contribution:** 2
**Rating:** 2
**Confidence:** 4

**Summary:**

This paper proposes a unified framework called Distribution-Aware Synergistic Evolution (DASE) that integrates discriminative and generative paradigms for few-shot learning. The key idea is to enable mutual reinforcement between a CLIP-based discriminator and a  Diffusion-based generator. The discriminative pathway refines class mean and covariance estimation by incorporating generated samples, while the generative pathway uses calibrated class prototypes from the discriminator as semantic anchors to enhance the fidelity of synthesized images. The framework is applied to three tasks—few-shot class-incremental learning (FSCIL), open-set recognition (OSR), and few-shot image generation (FSIG)—demonstrating moderate improvements in accuracy, AUROC, and FID on CUB200 and miniImageNet. Overall, the paper aims to bridge the gap between discrimination and generation in few-shot scenarios through iterative distribution calibration.

**Strengths:**

1. The paper’s structure and writing are clear, making the method easy to follow. Figures and tables are properly formatted, and the overall presentation is neat.
2. The research problem itself is meaningful and relevant. Exploring how discriminative and generative models can mutually benefit each other under few-shot conditions addresses a long-standing challenge in vision research, and the proposed bidirectional feedback idea has potential value for future work on hybrid learning frameworks.

**Weaknesses:**

1. The literature review is incomplete. Many recent works in few-shot image generation are not discussed, such as ADAM [1], RICK [2], and GenDA [3], as well as several key few-shot class-incremental learning methods from the past two years.
2. The experimental design is limited. Most comparisons are only comparing with the variants of the proposed method. For example, in Table 3, the method is evaluated on few-shot image generation but without comparing to any existing FSIG methods. Furthermore, well-known diffusion-based personalization methods like DreamBooth and Textual Inversion are mentioned in the related work but not included in experiments. This is important, since both CLIP and Stable Diffusion are pre-trained on massive paired datasets, and their inherent generalization ability could overshadow the claimed contribution of the proposed method.
3. The analysis of the experimental results is not convincing. In Figure 2, the visual differences between the proposed method and the baseline are not significant; some images appear almost identical, and only a few qualitative examples are shown, making the analysis not statistically supported.
4. There are minor typographical errors. For instance, the abstract ends with “few-shot incremental generation (FSIG),” but based on the paper’s context, it should be “few-shot image generation.”

[1] Few-shot Image Generation via Adaptation-Aware Kernel Modulation

[2] Exploring Incompatible Knowledge Transfer in Few-shot Image Generation

[3] FEW-SHOT CROSS-DOMAIN IMAGE GENERATION VIA INFERENCE-TIME LATENT-CODE LEARNING

**Questions:**

Please refer to Weaknesses.

---

> ### Author Response · Authors · 2025-11-28
>
> We appreciate the reviewer's constructive comments and have provided a point-by-point response to each, as detailed below.
>
> > 1. The literature review is incomplete. Many recent works in few-shot image generation are not discussed, such as ADAM, RICK, and GenDA, as well as several key few-shot class-incremental learning methods from the past two years.
>
> We sincerely thank the reviewer for pointing out these omissions. We have now expanded the literature review in the main body of our paper to include a comprehensive discussion of recent advancements in both few-shot image generation and few-shot class-incremental learning. Please refer to the highlighted sections in the updated RELATED WORK section of our manuscript.
>
> > 2. The experimental design is limited. Most comparisons are only comparing with the variants of the proposed method. For example, in Table 3, the method is evaluated on few-shot image generation but without comparing to any existing FSIG methods. Furthermore, well-known diffusion-based personalization methods like DreamBooth and Textual Inversion are mentioned in the related work but not included in experiments. This is important, since both CLIP and Stable Diffusion are pre-trained on massive paired datasets, and their inherent generalization ability could overshadow the claimed contribution of the proposed method.
>
> We thank the reviewer for this suggestion and have now included comparisons with recent state-of-the-art method DreamBooth in both class accuracy (ACC) and generation quality (FID) for the few-shot image generation task:
>
> | Method         | Backbone | CUB200 |       |
> | -------------- | -------- | ------ | ----- |
> |                |          | ACC ↑  | FID ↓ |
> | Baseline       | CLIP-L   | 77.82  | 20.66 |
> | DreamBooth     | CLIP-L   | 75.23  | 28.98 |
> | DASE(Phase I)  | CLIP-L   | 77.98  | 20.44 |
> | DASE(Phase II) | CLIP-L   | 78.12  | 19.44 |
>
> As shown in the table below, our method achieves both superior FID (19.44 vs. 28.98) and higher classification accuracy (78.12 vs. 75.23) on CUB-200. We attribute this to a key difference: DreamBooth excels at reproducing specific subjects with detailed prompts but struggles with diversity under uniform prompts. Our method, guided by calibrated visual prototypes, better maintains class fidelity while ensuring diversity.
>
> > 3. The analysis of the experimental results is not convincing. In Figure 2, the visual differences between the proposed method and the baseline are not significant; some images appear almost identical, and only a few qualitative examples are shown, making the analysis not statistically supported.
>
> Regarding the concern about the limited number of qualitative examples, we have added more representative samples in the revised manuscript to provide a more comprehensive and statistically supported visual comparison. As highlighted by the yellow boxes in Figure 2, both the fine-grained class attributes (e.g., the white neck feathers of a White-necked Crow) and the structural semantic features (e.g., the beak shape and overall morphology of the bird) show progressive improvement in the generated samples.
>
> > 4. There are minor typographical errors. For instance, the abstract ends with “few-shot incremental generation (FSIG),” but based on the paper’s context, it should be “few-shot image generation.”
>
> We sincerely thank the reviewer for catching these typographical errors and have changed "few-shot incremental generation (FSIG)" to "few-shot image generation (FSIG)" in the abstract.

---

### Official Review · Reviewer_rDSf · 2025-11-01

**Soundness:** 2
**Presentation:** 2
**Contribution:** 2
**Rating:** 4
**Confidence:** 3

**Summary:**

This paper introduces the Distribution-Aware Synergistic Evolution (DASE), an approach designed to unify few-shot discrimination and generation tasks. The core idea is to establish a bidirectional, mutually beneficial relationship between a discriminative and a generative pathway. The framework operates in two phases: (I) The discriminative model calibrates the prototype for new few-shot classes, which then serves as a semantic anchor to guide the generative model in producing higher-fidelity images. (II) The discriminative model, in turn, leverages the synthetic samples from the generator to estimate a more robust feature covariance matrix, addressing the unreliability of covariance estimation from scarce data. Experiments demonstrate that this synergistic loop enhances performance across Few-Shot Class-Incremental Learning, Open-Set Recognition, and Few-Shot Image Generation on the CUB-200 and miniImageNet datasets.

**Strengths:**

The proposed framework is technically plausible and is supported by empirical validation.

**Weaknesses:**

1. The abstract makes a strong assertion that “discriminative models are better at estimating the class center, while generative models are better at modeling the data variance”. However, this claim lacks sufficient theoretical justification or citations from prior literature.
2. The method section does not present any explicit loss or objective function; the algorithm is only described procedurally, which weakens its mathematical rigor.
3. Experiments are conducted only on miniImageNet and CUB-200, lacking more challenging benchmarks, which limits the assessment of generalization.
4. The paper omits comparisons with recent state-of-the-art methods in open-set recognition and few-shot image generation, making it difficult to evaluate the competitiveness of the proposed approach.
5. The method introduces several critical hyperparameters, yet the paper provides no discussion or sensitivity analysis regarding their selection, which reduces reproducibility.

**Questions:**

1. The paper states that the calibrated visual prototype $\mu_c$ is injected into Stable Diffusion by replacing the $t_{EOS}$ and $t_{PAD}$ embeddings in the CLIP text encoder. This is a rather unusual and insufficiently justified design choice. The authors should provide ablation studies or theoretical reasoning to support this key component.
2. In Section 5.1, the authors mention using a “class-specific prior preservation loss” during LoRA training on the CUB-200 dataset, but not on miniImageNet. Why is this regularization applied to only one dataset? This inconsistency makes the comparison between datasets unfair and undermines the claimed generality of the method.
3. In Table 1, the $A_{last}$ score of DASE (Phase I) on CUB-200 is reported as 78.51%, whereas in Table 4, the corresponding method (“Calibrated Class Distribution”) achieves 78.15%. These two entries appear to represent the same experiment (the discriminative Phase I only), yet the results are inconsistent. Please clarify this discrepancy.

---

> ### Author Response · Authors · 2025-11-28
>
> We appreciate the reviewer's constructive comments and have provided a point-by-point response to each, as detailed below.
>
> > 1. The abstract makes a strong assertion that “discriminative models are better at estimating the class center, while generative models are better at modeling the data variance”. However, this claim lacks sufficient theoretical justification or citations from prior literature.
>
> We thank the reviewer for this insightful comment and find the assertion in the abstract lacked sufficient contextualization. To clarify, our claim is specifically intended within the context of **few-shot learning scenarios**. Therefore, we have rephrased the statement in the manuscript to: "In few-shot scenarios, prototype-based discriminative methods are better at estimating the class center, while generative models are better at modeling the data variance."
>
> This revised formulation more accurately reflects the specific strengths of each model type, as evidenced by prior literature and our methodological design:
>
> **For Discriminative Models:**
>
> Discriminative models excel at learning decision boundaries by focusing on the conditional probability P(Y|X) and enhance **inter-class separability**.[1] In the context of prototype-based methods, this translates to a more accurate estimation of class centroids[2].  This principle is foundational to many few-shot learning algorithms, most notably **Prototypical Networks**[2], which explicitly leverage class centers for classification . The discriminative pathway in our framework is built directly upon this established principle.
>
> **For Generative Models:**
>
> Generative models aim to capture the underlying data distribution P(X) or P(X|Y). To generate realistic and diverse samples, they must capture the **intrinsic similarities among samples from the same class and model the complete data variance** [3]. This capability is empirically demonstrated by the success of various deep generative models, such as GANs and Diffusion Models, which are designed to learn complex, high-dimensional data distributions and have been shown to effectively **capture the feature distribution** of classes, even from limited data . Our use of a generative component to "expand" the estimated distribution for few-shot classes is designed to leverage this inherent capability of modeling detailed data variance.
>
> ---
>
> **References**
>
> 1. Hastie, Trevor. "The elements of statistical learning: data mining, inference, and prediction." (2009).
> 2. Snell, Jake, Kevin Swersky, and Richard Zemel. "Prototypical networks for few-shot learning." *Advances in neural information processing systems* 30 (2017).
> 3. Goodfellow, Ian J., et al. "Generative adversarial nets." *Advances in neural information processing systems* 27 (2014).
>
>
>
> > 2. The method section does not present any explicit loss or objective function; the algorithm is only described procedurally, which weakens its mathematical rigor.
>
> We thank the reviewer for this observation and are happy to provide clarification. We have updated the manuscript accordingly, and we elaborate further here to explicitly explain the **training-free nature of the discriminative pathway** and to **detail the training objective of the generative pathway**.
>
> **The Discriminative Pathway does not involve trainable parameters or an explicit loss function.** We employ a frozen backbone to extract features and perform classification and open-set recognition based on the estimated feature distributions. The backbone remains fixed to prevent catastrophic forgetting.
>
> **In the Generative Pathway，we fine-tune a pretrained text-to-image diffusion model using low-rank adaptation (LoRA).**  During fine-tuning, the original model parameters $\theta_0$ remain frozen, and only the LoRA parameters $\phi$ are updated. The denoising network with LoRA adaptation is denoted by $\epsilon_{\theta_0,\phi}$. The fine-tuning objective follows the standard noise-prediction loss used in diffusion models:
>
> $$
> L = E_{(x,t),\epsilon,s} [ || \epsilon - \epsilon_{\theta_0,\phi}(x_s, s, T') ||_2^2 ].
> $$
>
> Here, $x_s$ is the noisy sample constructed at timestep $s$, $\epsilon$ is Gaussian noise, and $T'$ denotes the text embedding produced by our custom text encoder. In this formulation, $\theta_0$ represents the frozen pretrained diffusion parameters, while $\phi$ denotes the trainable low-rank matrices introduced by LoRA.

---

> > ### Author Response · Authors · 2025-11-28
> >
> > > 3. Experiments are conducted only on miniImageNet and CUB-200, lacking more challenging benchmarks, which limits the assessment of generalization.
> >
> > We agree with the reviewer that evaluation on more challenging benchmarks would strengthen our work. We initially considered classic benchmarks including CIFAR-100 for FSCIL and more complex datasets like iNaturalist. However, CIFAR-100's low image resolution (32×32) poses significant challenges for evaluating generation quality, which would compromise the meaningfulness of our generative results. We acknowledge this limitation and will include experiments on more diverse and challenging benchmarks in future work to further validate the generalization capability of our approach.
> >
> >
> >
> > > 4. The paper omits comparisons with recent state-of-the-art methods in open-set recognition and few-shot image generation, making it difficult to evaluate the competitiveness of the proposed approach.
> >
> > We thank the reviewer for this valuable suggestion. We have now included comprehensive comparisons with recent state-of-the-art methods in both open-set recognition (OSR) and few-shot image generation (FSIG).
> >
> > **Open-Set Recognition Comparisons:**
> >
> > We evaluated our method against two recent OSR approaches: **MGPL**[1], which employs a more complex multi-Gaussian prototype modeling per class, and **GHOST**[2], which uses multivariate Gaussian distributions with diagonal covariance similar to our baseline. Under the same experimental setup as in Table 2 of our paper (5-shot distribution modeling for each class in every incremental session), the results (AUROC %) are as follows:
> >
> > | Method         | Backbone | miniImageNet | Cub200 |
> > | -------------- | -------- | ------------ | ------ |
> > | MGPL           | CLIP-L   | 26.79        | 21.42  |
> > | GHOST          | CLIP-L   | 53.37        | 44.67  |
> > | Baseline       | CLIP-L   | 50.19        | 42.77  |
> > | Ours(Phase I)  | CLIP-L   | 54.23        | 45.78  |
> > | Ours(Phase II) | CLIP-L   | 55.34        | 47.08  |
> >
> > The results demonstrate the effectiveness of our approach in few-shot settings. While MGPL's complex model fails to achieve stable distribution estimation, GHOST maintains modeling stability and surpasses our baseline. Our method, through synergistic evolution between generation and discrimination, achieves more accurate class center and covariance estimation, yielding the best performance.
> >
> > **few-shot image generation Comparisons:**
> >
> > We have included comparisons with recent state-of-the-art method DreamBooth in both class accuracy (ACC) and generation quality (FID) for the few-shot image generation task:
> >
> > | Method         | Backbone | CUB200 |       |
> > | -------------- | -------- | ------ | ----- |
> > |                |          | ACC ↑  | FID ↓ |
> > | Baseline       | CLIP-L   | 77.82  | 20.66 |
> > | DreamBooth     | CLIP-L   | 75.23  | 28.98 |
> > | DASE(Phase I)  | CLIP-L   | 77.98  | 20.44 |
> > | DASE(Phase II) | CLIP-L   | 78.12  | 19.44 |
> >
> > As shown in the table below, our method achieves both superior FID (19.44 vs. 28.98) and higher classification accuracy (78.12 vs. 75.23) on CUB-200. We attribute this to a key difference: DreamBooth excels at reproducing specific subjects with detailed prompts but struggles with diversity under uniform prompts. Our method, guided by calibrated visual prototypes, better maintains class fidelity while ensuring diversity.
> >
> > ---
> >
> > **References**
> >
> > 1. Liu, Jiaming, et al. "Learning multiple gaussian prototypes for open-set recognition." *Information Sciences* 626 (2023): 738-753.
> >
> > 2. Rabinowitz, Ryan, et al. "GHOST: Gaussian Hypothesis Open-Set Technique." *Proceedings of the AAAI Conference on Artificial Intelligence*. Vol. 39. No. 6. 2025.

---

> > > ### Author Response · Authors · 2025-11-28
> > >
> > > > 5. The method introduces several critical hyperparameters, yet the paper provides no discussion or sensitivity analysis regarding their selection, which reduces reproducibility.
> > >
> > > We appreciate this important point and now provide a comprehensive sensitivity analysis for three key hyperparameters:
> > >
> > > - $\lambda$  (Eq. 4 & 5): Controls base-novel knowledge transfer
> > > - $\eta$  (Eq. 4 & 5): Provides numerical stability
> > > - $\tau$  : Confidence threshold for filtering generated samples in Discrimination Phase II
> > >
> > > The experimental results on CUB200 (AUROC) are as follows:
> > >
> > > | $\lambda$      | 0     | 0.25  | 0.5   | 0.75  | 1     |
> > > | -------------- | ----- | ----- | ----- | ----- | ----- |
> > > | Baseline       | 42.23 | 42.46 | 42.77 | 42.31 | 41.98 |
> > > | Ours(Phase I)  | 45.27 | 45.59 | 45.78 | 45.27 | 44.84 |
> > > | Ours(Phase II) | 46.45 | 46.93 | 47.08 | 46.63 | 46.22 |
> > >
> > > The optimal performance is achieved at $\lambda$=0.5, indicating a balanced integration of base and novel class knowledge. Values deviating from this optimum lead to performance degradation, either by over-relying on base class statistics (low $\lambda$) or overfitting to few-shot samples (high $\lambda$).
> > >
> > > | $\eta$         | 0     | 0.1   | 0.2   | 0.5   | 1     |
> > > | -------------- | ----- | ----- | ----- | ----- | ----- |
> > > | Baseline       | 42.66 | 42.74 | 42.77 | 42.7  | 42.65 |
> > > | Ours(Phase I)  | 45.68 | 45.74 | 45.78 | 45.72 | 45.68 |
> > > | Ours(Phase II) | 47.01 | 47.05 | 47.08 | 47.01 | 46.98 |
> > >
> > > The results show relative insensitivity to η within the tested range, with optimal performance at η=0.2. This confirms that our method maintains numerical stability without heavy reliance on this regularization parameter.
> > >
> > > | $\tau$         | 0     | 0.25  | 0.5   | 0.75  | 1     |
> > > | -------------- | ----- | ----- | ----- | ----- | ----- |
> > > | Baseline       | 42.62 | 42.69 | 42.77 | 42.72 | 42.64 |
> > > | Ours(Phase I)  | 45.66 | 45.72 | 45.78 | 45.74 | 45.68 |
> > > | Ours(Phase II) | 46.97 | 47.04 | 47.08 | 47.05 | 47.01 |
> > >
> > > The optimal threshold is $\tau$=0.5, effectively filtering out low-confidence generated samples while retaining sufficient data for covariance refinement. This demonstrates the importance of selective sample utilization in our framework.
> > >
> > > > 6. The paper states that the calibrated visual prototype $\mu_c$ is injected into Stable Diffusion by replacing the $t_{EOS}$ and $t_{PAD}$ embeddings in the CLIP text encoder. This is a rather unusual and insufficiently justified design choice. The authors should provide ablation studies or theoretical reasoning to support this key component.
> > >
> > > We appreciate the opportunity to clarify this design choice. Our approach has feasible precedents in existing researchs [1] [2], and our design choice of replacing specific token embeddings in the CLIP text encoder with a calibrated visual prototype can be regarded as an extension of this general mechanism. The clip2latent [1] method exploits the alignment between CLIP image and text embeddings as a general bridge for connecting language and vision signals with generative models. In addition, the Brush2Prompt[2] method presents a related paradigm by applying CLIP image embeddings within a text-guided generation framework in Stable Diffusion. By injecting visual representations into components originally designed for textual embeddings, this method also crosses the modality boundary between visual features and text embeddings. These works collectively indicate that using visual embeddings to replace or complement text embeddings within the text-encoder pathway is already a feasible and meaningful practice.
> > >
> > > ---
> > >
> > > **References**
> > >
> > > 1. Pinkney, Justin NM, and Chuan Li. "clip2latent: Text driven sampling of a pre-trained stylegan using denoising diffusion and clip." *arXiv preprint arXiv:2210.02347* (2022).
> > > 2. Chiu, Mang Tik, et al. "Brush2Prompt: Contextual Prompt Generator for Object Inpainting." *Proceedings of the IEEE/CVF Conference on Computer Vision and Pattern Recognition*. 2024.

---

> > > > ### Author Response · Authors · 2025-11-28
> > > >
> > > > > 7. In Section 5.1, the authors mention using a “class-specific prior preservation loss” during LoRA training on the CUB-200 dataset, but not on miniImageNet. Why is this regularization applied to only one dataset? This inconsistency makes the comparison between datasets unfair and undermines the claimed generality of the method.
> > > >
> > > > This is a valid concern, and we appreciate the opportunity to clarify. The application of the class-specific prior preservation loss solely to CUB-200 is a deliberate, dataset-specific design choice, but it does not compromise the fairness of our evaluation.
> > > >
> > > > The key point is that **all comparisons between different methods on the same dataset are conducted under identical conditions**, which ensures fair within-dataset comparisons. Since our method demonstrates effectiveness on both datasets (as shown in Tables 3), its generalization capability is well validated across different data characteristics.
> > > >
> > > > The distinction in applying this specific loss stems from fundamental dataset properties. The class-specific prior preservation loss requires a dataset with a coherent semantic domain to construct an effective prior. CUB-200, with all classes belonging to the unified concept of "birds," perfectly satisfies this requirement, making the regularization beneficial for stabilizing fine-grained LoRA training. In contrast, miniImageNet's diverse semantic categories (spanning animals, tools, and various objects) lack a unified domain-level prior that would make such regularization effective or meaningful.
> > > >
> > > >
> > > >
> > > > > 8. In Table 1, the score of DASE (Phase I) on CUB-200 is reported as 78.51%, whereas in Table 4, the corresponding method (“Calibrated Class Distribution”) achieves 78.15%. These two entries appear to represent the same experiment (the discriminative Phase I only), yet the results are inconsistent. Please clarify this discrepancy.
> > > >
> > > > We thank the reviewer for catching this discrepancy. The value of 78.15% in Table 4 was indeed a typographical error. The correct value should be 78.51%, consistent with Table 1. We sincerely apologize for this oversight and will correct it in the final version.

---

### Author Response · Authors · 2025-12-02
**Consolidated Response to Reviewer Comments: Strengths, Concerns, and Improvements Made**

Dear PCs, SACs, ACs and reviewers,

In light of the closure of further discussion on the platform, we provide below a consolidated summary for your reference. It outlines the strengths acknowledged by the reviewers, details their key concerns and questions, and presents our corresponding responses, clarifications, and improvements.

### 1. Summary of Strengths Acknowledged by Reviewers

The reviewers collectively recognized several key strengths of our work:

*   **Novelty and a Unified Framework (rDSf,iTVM,nbJQ,7Td4):** Multiple reviewers highlighted the **novelty of the bidirectional interaction** between discriminative and generative models. They found the proposed novel(**7Td4**), unified(**nbJQ,7Td4**), distribution-aware framework to be “technically plausible”(**rDSf**) and meaningful for addressing the longstanding challenge of few-shot learning(**iTVM**).
*   **Strong Empirical Validation:** The “rigorous experimental design”(**7Td4**) was commended, with thorough evaluations across three tasks—Few-Shot Class-Incremental Learning, Open-Set Recognition, and Few-Shot Image Generation—alongside comprehensive ablation studies validating each component.
*   **Clear Presentation:** The manuscript was praised for its **clarity in writing and presentation**, with well-structured arguments and “properly formatted”(**iTVM**) figures/tables, making the method “easy to follow”(**iTVM**).
*   **Significant Research Direction:** The core problem of enabling generative and discriminative models to benefit each other under data scarcity was “meaningful and relevant”(**iTVM**) for future hybrid learning paradigms.

### 2. Summary of Key Concerns and Our Responses/Improvements

We have provided detailed point-by-point responses to all reviewer concerns. The major issues and our corresponding actions are summarized below:

*   **1. Theoretical Justification:**
    *   **Concern:** (**rDSf,7Td4**) The strong claim in the abstract that "**discriminative models are better at estimating the class center, while generative models are better at modeling the data variance**" lacked sufficient theoretical backing.
    *   **Response & Improvement:** We have revised the statement to be more precise: **"In few-shot scenarios, prototype-based discriminative methods are better at estimating the class center, while generative models are better at modeling the data variance."** We provided theoretical grounding by referencing foundational literature.
*   **2. Limited Experimental Scope and Comparisons:**
    *   **Concerns:**
        *   (**rDSf**) Experiments were limited to miniImageNet and CUB-200.
        *   (**rDSf,iTVM**) Lacked comparisons with recent state-of-the-art (SOTA) methods in OSR and FSIG.
        *   (**iTVM**) Literature review was incomplete regarding recent FSIG and FSCIL works.
        *   (**7Td4**) No analysis of performance under domain shift for OSR
    *   **Response & Improvement:**
        *   **We have now included extensive comparisons with SOTA methods:**
            *   For OSR: Compared with **MGPL** and **GHOST** (see rebuttal to Reviewer rDSf, Table added).
            *   For FSIG: Compared with **DreamBooth** (see rebuttal to Reviewers rDSf & iTVM, Tables added).
        *   The **Related Work section has been expanded** to discuss recent advances, including methods like ADAM, RICK, and GenDA.
        *   We **added a cross-domain OSR evaluation** (training on CUB200, testing on iNaturalist and CIFAR-100 as OOD). Our method showed consistent improvements over the baseline, indicating enhanced robustness (see rebuttal to Reviewers nbJQ & 7Td4, Table added).
*   **3. Lack of Methodological Detail and Hyperparameter Analysis:**
    *   **Concerns:**
        *   (**rDSf**) No explicit loss function was provided for the method section.
        *   (**rDSf**) No sensitivity analysis for critical hyperparameters.
        *   (**rDSf**) Insufficient justification for key design choices (e.g., prototype injection into CLIP text encoder).
    *   **Response & Improvement:**
        *   We clarified the **training-free nature of the discriminative pathway** and explicitly provided the **noise-prediction loss for the generative LoRA fine-tuning**.
        *   We provided a **comprehensive sensitivity analysis** for hyperparameters λ, η, and τ (see rebuttal to Reviewer rDSf, Tables added), demonstrating robustness and optimal settings.
        *   We justified the prototype injection design by referencing related works that bridge visual and textual embeddings in diffusion models (e.g., clip2latent, Brush2Prompt).

---

> ### Author Response · Authors · 2025-12-02
>
> *   **4. Oversimplification with Diagonal Covariance:**
>     *   **Concern:** (**nbJQ,7Td4**) The diagonal Gaussian assumption may be too simplistic.
>     *   **Response & Improvement:** We conducted an **ablation study comparing full, low-rank (PPCA, block-diagonal), and diagonal covariance matrices**. Results confirmed that in the extreme low-data regime (5-shot), diagonal covariance offers the best stability-performance trade-off. Full covariance failed due to being severely underdetermined (see rebuttal to Reviewers nbJQ & 7Td4, Table added).
>
> ### 3. Conclusion
>
> In conclusion, we have addressed all substantive concerns raised by the reviewers through clarifications, additional theoretical justification, expanded experiments and comparisons, and detailed ablation/sensitivity analyses. We believe these revisions have effectively strengthened the manuscript's rigor, completeness, and persuasive power. We are grateful for the reviewers' insightful feedback and are prepared to incorporate all these improvements into the final version.

---

### Meta-Review · Area_Chair_wgjU · 2026-01-13

**Summary:**

The reviews are split (two 4s, one 2, one 6) with none championing the paper. The strongest negative review questions whether the incremental gains are convincingly attributable to the proposed synergy or to leveraging strong pretrained CLIP/Stable Diffusion.

Across reviewers, the main concerns were regarding (1) empirical completeness/competitiveness - initial submission lacked comparisons to recent OSR and FSIG baselines and had limited benchmark scope (only miniImageNet/CUB), making it hard to judge whether DASE is competitive beyond internal ablations; (2) methodological rigor/clarity - missing explicit objective(s) in the method, unclear/under-justified design choices (notably prototype injection into CLIP text encoder), and missing hyperparameter sensitivity initially reduced confidence and reproducibility; (3) strength of evidence for qualitative generation claims - the qualitative analysis are strong enough, raising questions about how robust the FSIG improvements are; and (4) modeling assumptions/generalization - diagonal covariance modeling may be simplistic, and robustness under domain shift and broader backbone/diffusion-model generality were initially not demonstrated.

**Reviewer Concerns:**

Some concerns were largely addressed, such as

- Missing loss/objective detail (rDSf): Authors clarified “training-free” discriminative pathway and provided the diffusion noise-prediction loss for LoRA fine-tuning.

- Hyperparameter sensitivity (rDSf, nbJQ): Added sensitivity analyses for key parameters (\lambda/\eta/\tau), showing relative robustness and identifying reasonable defaults.

- Comparisons to OSR/FSIG baselines (rDSf, iTVM): Added OSR comparisons (MGPL, GHOST) and FSIG comparison (DreamBooth), with reported improvements.

- Diagonal covariance vs. richer structure (nbJQ, 7Td4): Added ablations (full/PPCA/block-diagonal/diagonal), supporting the claim that full covariance is unstable in 5-shot and diagonal/low-rank are similar.

- Domain-shift OSR (7Td4, nbJQ): Added cross-domain OSR evaluation (e.g., CUB->iNaturalist/CIFAR-100) showing consistent gains.

However, some concerns still lingered, such as

- Scope/generalization beyond the current instantiation (7Td4): The rebuttal largely defers "larger diffusion models/non-CLIP backbones" to future work. That’s acceptable, but it weakens any implication of broad model-agnostic generality—especially since the discriminator->generator bridge relies on CLIP alignment.

- Strength of the FSIG evidence (iTVM): Adding more qualitative samples helps, but the core critique (visual differences not clearly compelling; limited statistical support) remains only partially resolved unless the revision adds more systematic human/automatic evaluations (diversity, fidelity, prompt adherence) beyond a small set of examples.

- Prototype injection justification (rDSf): Citing related mechanisms (clip2latent/Brush2Prompt) is helpful, but it's still not a fully satisfying justification without a direct ablation demonstrating that this specific token replacement strategy is necessary/better than simpler alternatives (e.g., prompt tuning, textual inversion-style embeddings, adapter layers) under matched compute.

- Benchmark breadth (rDSf): The rebuttal explains why CIFAR-100 is awkward for generation, but the overall evaluation is still narrow; "future work" is reasonable, yet it remains a weakness.

**Reviewer Scores:**

Reviewer rDSf (score 4 -> likely 5): The rebuttal directly addresses most of rDSf’s concrete issues: objective, hyperparameter sensitivity, missing OSR/FSIG baselines, prototype-injection justification (at least with precedent), and the table typo. Remaining concern is benchmark breadth and lingering unease about the injection choice. Net: small upward move to a weak accept / borderline.

Reviewer iTVM (score 2 -> likely 3, possibly 4 in best case): iTVM’s main objections were incomplete related work, lack of FSIG baselines, and unconvincing qualitative evidence. Related work and DreamBooth comparison are addressed, and more qualitative samples were added, but the “pretraining could overshadow contribution” skepticism may persist. Net: likely modest increase to 3 (still reject), unless the added evidence in the revision is unusually strong and clearly isolates the synergy benefit.

Reviewer nbJQ (score 4 -> likely 5): nbJQ asked for (i) richer covariance modeling tradeoffs, (ii) sensitivity to filtering quality, (iii) ablations isolating the synergy directions, and (iv) some generalization evidence. The rebuttal supplies covariance ablations, sensitivity, explicit Phase I vs II gains, and cross-domain OSR.

Reviewer 7Td4 (score 6 -> likely 6, maybe 7): 7Td4 already leans accept but flagged covariance structure and domain shift; both are addressed with ablations and cross-domain OSR. The "larger diffusion/non-CLIP" question remains future work.

However, even with the potentially updated scores, the paper remains borderline. In addition, the lingering concerns about generality and strength of FSIG evidence make it difficult to recommend acceptance.

---

### Decision · Program_Chairs · 2026-01-26

Reject